# Pseudo-Poisson Distributions with Concomitant Variables

**Barry C. Arnold** [1,*] **and Bangalore G. Manjunath** [2,*]

1    Department of Statistics, University of California, Riverside, CA 92521, USA
2    School of Mathematics and Statistics, University of Hyderabad, Hyderabad 500046, India
*    Correspondence: barnold@ucr.edu (B.C.A.); bgmanjunath@gmail.com (B.G.M.)

**Abstract:** It has been argued in Arnold and Manjunath (2021) that the bivariate pseudo-Poisson distribution will be the model of choice for bivariate data with one equidispersed marginal and the other marginal over-dispersed. This is due to its simple structure, straightforward parameter estimation and fast computation. In the current note, we introduce the effects of concomitant variables on the bivariate pseudo-Poisson parameters and explore the distributional and inferential aspects of the augmented models. We also include a small simulation study and an example of application to real-life data.

**Keywords:** correlation; likelihood ratio test; maximum likelihood estimators; pseudo-Poisson; regression

## 1. Introduction

The classical "one-dimensional" Poisson distribution has historically been found to be useful in modeling a wide variety of "integer-valued" phenomena, such as the number of accidents and associated fatalities, disease advances, rate of rare event occurrences and so on. With regard to the Poisson model with concomitants, i.e., Poisson regression or count regression, its best known applications are in (*i*) modeling counts of bacteria exposed to various environmental conditions and dilutions; (*ii*) modeling counts of infant mortality among groups with different demographics. All these examples are typically modeled under the assumption of equi-dispersion. However, count data also exhibits over and under dispersion. In this context, the one-dimensional Conway–Maxwell–Poisson model or its regression version fills the bill precisely by allowing us to model over, equi- and under-dispersion data.

In general, bivariate count data, along with having marginal over-, under- or equi-dispersion, will also exhibit a variety of dependence structures. In particular, for linear dependence, the possible relations are positive or negative correlation. The classical bivariate Poisson model is appropriate for data having equi-dispersed marginals with positive correlation. Here again, the bivariate Conway–Maxwell–Poisson is more flexible in that it can adapt to both under and over dispersed data, see Sellers et al. [1]. Concerning bivariate Poisson regression models, the first version involving explanatory variables acting on the marginal means was introduced in Kocherlakota and Kocherlakota [2] based on the classical bivariate Poisson model. In addition, the derivation of Wald, score and likelihood ratio test statistics for testing a single coefficient parameter vector are discussed in Riggs et al. [3]. Zamani et al. [4] proposed a bivariate Poisson model which can be fitted to both positive and negatively correlated data. Recently, Chowdhury et al. [5] considered the Poisson–Poisson regression model (which is the particular case of the bivariate pseudo-Poisson model) to analyze the impact of covariates on the daily new cases and fatalities associated with the COVID-19 pandemic. Finally, we refer to Karlis and Ntzoufras [6] and the R package *bivpois* for maximum likelihood estimation, using an Expectation-Maximization (EM) algorithm, for diagonally inflated bivariate Poisson regression models.

In recent work, Arnold and Manjunath [7] recommended the bivariate pseudo-Poisson model to fit data which have one marginal and other conditional of the Poisson form. Due to its straightforward structure with no restrictions on the conditional mean function, it allows us to include a variety of dependence structures, including positive and negative correlation. In the following, we introduce explanatory variables acting on the pseudo-Poisson parameters. Thanks to the simple structure, the concomitant effects can be introduced into each of the parameters to generate a family of models with a variety of dependence structures. We refer to Arnold et al. [8] and Veeranna et al. [9] on Bayesian and goodness-of-fit tests for the bivariate pseudo-Poisson model, respectively, which can also be adapted to accommodate the presence of concomitant variables. We refer to Arnold et al. [10] and Filus et al. [11] for further reading on conditional specified models and the triangular transformations, respectively. Finally, we refer Ghosh et al. [12] on the recent results on bivariate count model which has both conditionals with a Poisson structure.

We next review the concept of multivariate pseudo-Poisson distributions, as discussed in Arnold and Manjunath [7].

**Definition 1.** *A $k$-dimensional random variable $\underline{X} = (X_1, X_2, \ldots, X_k)$ is said to have a $k$-dimensional pseudo-Poisson distribution if there exists a positive constant $\lambda_1$ such that*

$$X_1 \sim \mathscr{P}(\lambda_1)$$

*and $k-1$ functions $\{\lambda_\ell : \ell = 2, 3, \ldots, k\}$ where, for each $\ell$, $\lambda_\ell : \{0, 1, 2, \ldots\}^{(\ell-1)} \to (0, \infty)$ such that*

$$X_\ell | \underline{X}_{(\ell-1)} = \underline{x}_{(\ell-1)} \sim \mathscr{P}(\lambda_\ell(\underline{x}_{(\ell-1)})),$$

*where $\underline{X}_{(\ell-1)} = (X_1, \ldots, X_{l-1})^\top$. Note that there are no constraints on the forms of the functions $\lambda_\ell$, $\ell = 2, 3, \ldots, k$ that appear in the definition, save for measurability. In applications, it would typically be the case that the $\lambda_\ell$'s would be chosen to be relatively simple functions depending on a limited number of parameters.*

**Definition 2.** *A random pair of variables $(X_1, X_2)$ is said to have a bivariate pseudo-Poisson distribution if there exists a positive constant $\lambda_1$ such that*

$$X_1 \sim \mathscr{P}(\lambda_1)$$

*and a function $\lambda_2 : \{0, 1, 2, \ldots\} \to (0, \infty)$ such that, for every non-negative integer $x_1$,*

$$X_2 | X_1 = x_1 \sim \mathscr{P}(\lambda_2(x_1)).$$

The fact that there are no constraints on the $\lambda_2(x_1)$ allows us to adapt to a variety of dependence structures including positive or negative correlation.

**Example 1.** *A judicious choice of a parametric family for $\lambda_2(x_1)$ will admit positive and negative correlation between $X_1$ and $X_2$. For example, if we consider*

$$\lambda_2(x_1; \gamma, \delta) = 1 + (2\gamma - 1)(1 - e^{-\delta x_1}). \tag{1}$$

*For $\delta > 0$, the above function will be increasing if $\gamma > 1/2$, decreasing if $\gamma < 1/2$ and constant if $\gamma = 1/2$. Consequently, $X_1$ and $X_2$ will have a positive correlation if $\gamma > 1/2$, negative correlation if $\gamma < 1/2$ and will be uncorrelated if $\gamma = 1/2$. A more general model with the same properties can be obtained by replacing $1 - e^{-\delta x_1}$ by $F(x_1; \underline{\theta})$, a parameterized family of distribution functions with support $(0, \infty)$.*

## 2. Incorporating Concomitant Variables

In many (perhaps, most) applications, in addition to the observed values of the $(X_{1,i}, X_{2,i})$'s pairs, there will be available values of arrays of concomitant variables which are

expected to influence the stochastic behavior of the observed data points. A straightforward manner in which to incorporate vectors of concomitant variables $\underline{u}_i = (u_{1i}, \ldots, u_{di})^\top$ into the model is as follows:

$$X_1 \sim \mathscr{P}\left(\lambda_1 \exp\left(\underline{\alpha}^\top \underline{u}\right)\right) \tag{2}$$

and

$$X_2 | X_1 = x_1 \sim \mathscr{P}\left(\lambda_2 \exp\left(\underline{\beta}^\top \underline{u}\right) + \lambda_3 \exp\left(\underline{\gamma}^\top \underline{u}\right) x_1\right) \tag{3}$$

where $\lambda_1 > 0$, $\lambda_2 \geq 0$, $\lambda_3 > 0$, $\underline{\alpha} = (\alpha_1, \ldots, \alpha_d)^\top$, $\underline{\beta} = (\beta_1, \ldots, \beta_d)^\top$ and $\underline{\gamma} = (\gamma_1, \ldots, \gamma_d)^\top$ are $d$-dimensional unknown parameters.

There are certainly many other manners in which one can model the influence of concomitant variables. If there is scientific justification for alternative models that do not introduce the concomitants via log-linear adjustments of the form specified in (2) and (3), then one should certainly utilize the scientifically appropriate link functions.

Just as in classical multiple regression, it is worthwhile to determine whether a simple linear dependence assumption for the effect of concomitants will be adequate to fit the data. In the remainder of this paper, we will focus on the simple model (2) and (3).

### 3. Moments

In the following, we derive some population moments for the model specified in (2) and (3).

$$
\begin{aligned}
E(X_1) = Var(X_1) &= \lambda_1 \exp\left(\underline{\alpha}^\top \underline{u}\right) \\
E(X_2) &= \lambda_2 \exp\left(\underline{\beta}^\top \underline{u}\right) + \lambda_1 \lambda_3 \exp\left((\underline{\gamma} + \underline{\alpha})^\top \underline{u}\right) \\
V(X_2) &= \lambda_2 \exp\left(\underline{\beta}^\top \underline{u}\right) + \lambda_1 \lambda_3 \exp\left((\underline{\gamma} + \underline{\alpha})^\top \underline{u}\right) \\
&\quad + \lambda_1 \lambda_3^2 \exp\left((2\underline{\gamma} + \underline{\alpha})^\top \underline{u}\right) \\
Cov(X_1, X_2) &= \lambda_1 \lambda_3 \exp\left((\underline{\gamma} + \underline{\alpha})^\top \underline{u}\right).
\end{aligned}
$$

The marginal dispersion indices are

$$
DI(X_1) = \frac{Var(X_1)}{E(X_1)} = 1.
$$

$$
DI(X_2) = \frac{Var(X_2)}{E(X_2)} = 1 + \frac{\lambda_1 \lambda_3^2 \exp\left((2\underline{\gamma} + \underline{\alpha})^\top \underline{u}\right)}{\lambda_2 \exp\left(\underline{\beta}^\top \underline{u}\right) + \lambda_1 \lambda_3 \exp\left((\underline{\gamma} + \underline{\alpha})^\top \underline{u}\right)}.
$$

For $\lambda_2 = 0$

$$
DI(X_2) = 1 + \lambda_3 \exp\left(\underline{\gamma}^\top \underline{u}\right).
$$

Define

$$
\begin{aligned}
E(\underline{X}) &= (E(X_1), E(X_2))^\top \\
&= \left(\lambda_1 \exp\left(\underline{\alpha}^\top \underline{u}\right), \lambda_2 \exp\left(\underline{\beta}^\top \underline{u}\right) + \lambda_1 \lambda_3 \exp\left((\underline{\gamma} + \underline{\alpha})^\top \underline{u}\right)\right)^\top
\end{aligned}
$$

and

$$
Cov(\underline{X}) = \begin{bmatrix} \lambda_1 \exp\left(\underline{\alpha}^\top \underline{u}\right) & \lambda_1 \lambda_3 \exp\left((\underline{\gamma} + \underline{\alpha})^\top \underline{u}\right) \\ \lambda_1 \lambda_3 \exp\left((\underline{\gamma} + \underline{\alpha})^\top \underline{u}\right) & \lambda_2 \exp\left(\underline{\beta}^\top \underline{u}\right) + \lambda_1 \lambda_3 \exp\left((\underline{\gamma} + \underline{\alpha})^\top \underline{u}\right) + \\ & + \lambda_1 \lambda_3^2 \exp\left((2\underline{\gamma} + \underline{\alpha})^\top \underline{u}\right) \end{bmatrix}
$$

By using the definition given in the paper by Kokonendji and Puig [13] page 183, the bivariate Fisher index of dispersion is given by

$$
GDI(\underline{X}) = \frac{\sqrt{E(\underline{X})^\top} Cov(\underline{X}) \sqrt{E(\underline{X})}}{E(\underline{X})^\top E(\underline{X})}
$$

which is a case of over-disperson, cf. Arnold and Manjunath [7] page 2311 for the dispersion index proof for the bivariate pseudo-Poisson distribution.

## 4. Statistical Inference

In this section, we obtain maximum likelihood estimators (m.l.e.) of parameters $\lambda_1$, $\lambda_2$, $\lambda_3$, $\underline{\alpha}$, $\underline{\beta}$ and $\underline{\gamma}$. In addition, we construct the likelihood ratio test for the possible parallelism, coincidence and significance of each of the regression coefficients.

### 4.1. Estimation

Let $(X_{1i}, X_{2i})^\top$, $i = 1, 2, \ldots, n$ be a bivariate count sample from the pseudo-Poisson distribution (in Section 2) and let $\underline{u}_1, \ldots, \underline{u}_n$ be $d$-dimensional known covariates. Then, the log-likelihood function is

$$
\begin{aligned}
\log L &= -\lambda_1 \sum_{i=1}^{n} \exp\left(\underline{\alpha}^\top \underline{u}_i\right) + \sum_{i=1}^{n} x_{1i} \log\left(\lambda_1 \exp\left(\underline{\alpha}^\top \underline{u}_i\right)\right) \\
&\quad - \sum_{i=1}^{n} \left(\lambda_2 \exp\left(\underline{\beta}^\top \underline{u}_i\right) + \lambda_3 \exp\left(\underline{\gamma}^\top \underline{u}_i\right) x_{1i}\right) \\
&\quad + \sum_{i=1}^{n} x_{2i} \log\left(\lambda_2 \exp\left(\underline{\beta}^\top \underline{u}_i\right) + \lambda_3 \exp\left(\underline{\gamma}^\top \underline{u}_i\right) x_{1i}\right) \\
&\quad - \sum_{i=1}^{n} \log(x_{1i}! x_{2i}!).
\end{aligned} \tag{4}
$$

Partial differentiation with respect to each parameters $\lambda_1$, $\lambda_2$ and $\lambda_3$ and equating to zero gives

$$
-\sum_{i=1}^{n} \exp\left(\underline{\alpha}^\top \underline{u}_i\right) + \sum_{i=1}^{n} x_{1i} \frac{\exp\left(\underline{\alpha}^\top \underline{u}_i\right)}{\lambda_1 \exp\left(\underline{\alpha}^\top \underline{u}_i\right)} = 0 \tag{5}
$$

$$
-\sum_{i=1}^{n} \exp\left(\underline{\beta}^\top \underline{u}_i\right) + \sum_{i=1}^{n} x_{2i} \frac{\exp\left(\underline{\beta}^\top \underline{u}_i\right)}{\lambda_2 \exp\left(\underline{\beta}^\top \underline{u}_i\right) + \lambda_3 \exp\left(\underline{\gamma}^\top \underline{u}_i\right) x_{1i}} = 0 \tag{6}
$$

$$
-\sum_{i=1}^{n} x_{1i} \exp\left(\underline{\gamma}^\top \underline{u}_i\right) + \sum_{i=1}^{n} x_{1i} x_{2i} \frac{\exp\left(\underline{\gamma}^\top \underline{u}_i\right)}{\lambda_2 \exp\left(\underline{\beta}^\top \underline{u}_i\right) + \lambda_3 \exp\left(\underline{\gamma}^\top \underline{u}_i\right) x_{1i}} = 0. \tag{7}
$$

Now, taking partial derivatives of $\log L$ with respect to $\alpha_j$, $\beta_j$ and $\gamma_j$ for $j \in \{1, \ldots, d\}$ and equating to zero yields

$$-\lambda_1 \sum_{i=1}^{n} u_{ji} \exp\left(\underline{\alpha}^{\top} \underline{u}_i\right) + \sum_{i=1}^{n} x_{1i} u_{ji} = 0 \tag{8}$$

$$-\sum_{i=1}^{n} u_{ji} \exp\left(\underline{\beta}^{\top} \underline{u}_i\right) + \sum_{i=1}^{n} x_{2i} u_{ji} \frac{\exp\left(\underline{\beta}^{\top} \underline{u}_i\right)}{\lambda_2 \exp\left(\underline{\beta}^{\top} \underline{u}_i\right) + \lambda_3 \exp\left(\underline{\gamma}^{\top} \underline{u}_i\right) x_{1i}} = 0 \tag{9}$$

$$-\sum_{i=1}^{n} x_{1i} u_{ji} \exp\left(\underline{\gamma}^{\top} \underline{u}_i\right) + \sum_{i=1}^{n} x_{1i} x_{2i} u_{ji} \frac{\exp\left(\underline{\gamma}^{\top} \underline{u}_i\right)}{\lambda_2 \exp\left(\underline{\beta}^{\top} \underline{u}_i\right) + \lambda_3 \exp\left(\underline{\gamma}^{\top} \underline{u}_i\right) x_{1i}} = 0. \tag{10}$$

In particular, consider $d = 1$ and let $u_1, \ldots, u_n$ be the observed covariates. The likelihood equations from (5) to (10) simplify to become (with notation $\alpha_1 = \alpha$, $\beta_1 = \beta$ and $\gamma_1 = \gamma$)

$$\lambda_1 \sum_{i=1}^{n} \exp(\alpha u_i) = \sum_{i=1}^{n} x_{1i} \tag{11}$$

$$\sum_{i=1}^{n} \exp(u_i \beta) = \sum_{i=1}^{n} x_{2i} \frac{1}{\lambda_2 + \lambda_3 \exp(u_i(\gamma - \beta)) x_{1i}} \tag{12}$$

$$\sum_{i=1}^{n} x_{1i} \exp(u_i \gamma) = \sum_{i=1}^{n} x_{1i} x_{2i} \frac{1}{\lambda_2 \exp(u_i(\beta - \gamma)) + \lambda_3 x_{1i}}. \tag{13}$$

In the same way,

$$\lambda_1 \sum_{i=1}^{n} u_i \exp\left(\alpha u_i\right) = \sum_{i=1}^{n} x_{1i} u_i \tag{14}$$

$$\sum_{i=1}^{n} u_i \exp\left(\beta u_i\right) = \sum_{i=1}^{n} x_{2i} u_i \frac{1}{\lambda_2 + \lambda_3 x_{1i} \exp(u_i(\gamma - \beta))} \tag{15}$$

$$\sum_{i=1}^{n} x_{1i} u_i \exp\left(\gamma u_i\right) = \sum_{i=1}^{n} x_{1i} x_{2i} u_i \frac{1}{\lambda_2 \exp(u_i(\beta - \gamma)) + \lambda_3 x_{1i}}. \tag{16}$$

Note that the equations from (11) to (16) do not yield explicit expressions for the maximum likelihood estimates. However, one can use numerical methods to solve the system of six equations with six unknown parameters.

*4.2. Likelihood Ratio Test*

The general form of a generalized likelihood ratio test statistic is of the form

$$\Lambda = \frac{\sup_{\theta \in \Theta_0} L(\theta)}{\sup_{\theta \in \Theta} L(\theta)} \tag{17}$$

Here, $\Theta_0$ is a subset of $\Theta$ and we envision testing $H_0 : \theta \in \Theta_0$. We reject the null hypothesis for small values of $\Lambda$.

Now, for the bivariate pseudo-Poisson model, the natural parameter space under the full model is $\Theta = \{(\lambda_1, \lambda_2, \lambda_3, \underline{\alpha}, \underline{\beta}, \underline{\gamma})^{\top} : \lambda_1 \geq 0, \lambda_2 \geq 0, \lambda_3 \geq 0, \underline{\alpha} \in \mathbb{R}^d, \underline{\beta} \in \mathbb{R}^d, \underline{\gamma} \in \mathbb{R}^d\}$. The m.l.e.'s under the complete parameter space are obtained by taking partial differentiation of Equation (4) with respect to $\lambda_1, \lambda_2, \lambda_3, \underline{\alpha}, \underline{\beta}, \underline{\gamma}$ and equating to zero. We denote the obtained numerical solution m.l.e.'s by $\hat{\lambda}_1, \hat{\lambda}_2, \hat{\lambda}_3$ and $\underline{\hat{\alpha}}, \underline{\hat{\beta}}, \underline{\hat{\gamma}}$.

**Remark 1.** *We used the "maxLik" optimization function in R (in the package "maxLik") to obtain the m.l.e.'s by numerical solution. This function also allows us to use a different methods of optimization using algorithms such as Newton–Raphson, Broyden–Fletcher–Goldfarb–Shanno, Berndt–Hall–Hall–Hausman, Berndt–Hall–Hall–Hausman, Simulated Annealing, Conjugate Gradients and Nelder–Mead methods. In the current paper, we use the Newton–Raphson method to estimate parameters and to compute their standard errors.*

### 4.2.1. Testing $H_0 : \underline{\alpha} = \underline{\beta} = \underline{\gamma} = \underline{0}$

In the following, we will construct a likelihood ratio test for testing whether the observed concomitant does not affect the distribution of $(X_1, X_2)$. Under the null hypothesis, the natural parameter space is $\Theta_0 = \{(\lambda_1, \lambda_2, \lambda_3, \underline{\alpha}, \underline{\beta}, \underline{\gamma})^\top : \lambda_1 > 0, \lambda_2 \geq 0, \lambda_3 \geq 0, \underline{\alpha} = \underline{0}, \underline{\beta} = \underline{0}, \underline{\gamma} = \underline{0}\}$. Now, taking partial derivatives of Equation (4) with respect to each parameters $\lambda_1$, $\lambda_2$, $\lambda_3$ and equating to zero yields

$$-n + \frac{1}{\lambda_1} \sum_{i=1}^n X_{1i} = 0 \tag{18}$$

$$-n + \sum_{i=1}^n \frac{X_{2i}}{\lambda_2 + \lambda_3 X_{1i}} = 0 \tag{19}$$

$$-\sum_{i=1}^n X_{1i} + \sum_{i=1}^n \frac{X_{1i} X_{2i}}{\lambda_2 + \lambda_3 X_{1i}} = 0. \tag{20}$$

Equation (18) is readily solved, to obtain the m.l.e. for $\lambda_1$, namely, $\hat{\lambda}_1^* = \overline{X}_1$. The remaining two Equations (19) and (20) must be solved numerically to obtain $\hat{\lambda}_2^*$, $\hat{\lambda}_3^*$.

Now let $\hat{\lambda}_1, \hat{\lambda}_2, \hat{\lambda}_3$ and $\underline{\hat{\alpha}}, \underline{\hat{\beta}}, \underline{\hat{\gamma}}$ be the m.l.e. estimates on unrestricted space. Then, the likelihood (as defined in Equation (4)) ratio test statistic is

$$\Lambda_1 = \frac{L(\hat{\lambda}_1^*, \hat{\lambda}_2^*, \hat{\lambda}_3^*, \underline{0}, \underline{0}, \underline{0})}{L(\hat{\lambda}_1, \hat{\lambda}_2, \hat{\lambda}_3, \underline{\hat{\alpha}}, \underline{\hat{\beta}}, \underline{\hat{\gamma}})}. \tag{21}$$

If $n$ is large, then $-2\log(\Lambda_1)$ may be compared with a suitable $\chi^2_{3d}$ percentile in order to decide whether $H_0$ should be rejected or not.

### 4.2.2. Testing $H_0 : \underline{\alpha} = \underline{0}$

Here, we are testing that the observed concomitant does not affect the marginal distribution of $X_1$. Note that under the null hypothesis, the natural parameter space is $\Theta_0 = \{(\lambda_1, \lambda_2, \lambda_3, \underline{\alpha}, \underline{\beta}, \underline{\gamma})^\top : \lambda_1 > 0, \lambda_2 \geq 0, \lambda_3 \geq 0, \underline{\alpha} = \underline{0}, \underline{\beta} \in \mathbb{R}^d, \underline{\gamma} \in \mathbb{R}^d\}$. Now, again taking partial derivatives of Equation (4) with respect to parameters $\lambda_1, \lambda_2$, $\lambda_3$ & $\underline{\beta}$, $\underline{\gamma}$ and equating to zero gives m.l.e.'s, denoted by $\hat{\lambda}_1^*, \hat{\lambda}_2^*, \hat{\lambda}_3^*, \underline{\hat{\beta}}^*$ and $\underline{\hat{\gamma}}^*$, respectively. The likelihood ratio test statistic is

$$\Lambda_2 = \frac{L(\hat{\lambda}_1^*, \hat{\lambda}_2^*, \hat{\lambda}_3^*, \underline{0}, \underline{\hat{\beta}}^* \underline{\hat{\gamma}}^*)}{L(\hat{\lambda}_1, \hat{\lambda}_2, \hat{\lambda}_3, \underline{\hat{\alpha}}, \underline{\hat{\beta}}, \underline{\hat{\gamma}})} \tag{22}$$

If $n$ is large, then $-2\log(\Lambda_2)$ may be compared with a suitable $\chi^2_d$ percentile in order to decide whether $H_0$ should be rejected or not.

### 4.2.3. Testing $H_0 : \underline{\beta} = \underline{\gamma} = \underline{0}$

In this case, we are testing whether the observed concomitant does not affect the conditional distribution of $X_2$ given $X_1$. Under the null hypothesis, the natural parameter space is $\Theta_0 = \{(\lambda_1, \lambda_2, \lambda_3, \underline{\alpha}, \underline{\beta}, \underline{\gamma})^\top : \lambda_1 > 0, \lambda_2 \geq 0, \lambda_3 \geq 0, \underline{\alpha} \in \mathbb{R}^d, \underline{\beta} = \underline{0}, \underline{\gamma} = \underline{0}\}$. Again, taking partial derivatives of Equation (4) with respect to each parameters $\lambda_1$, $\lambda_2$, $\lambda_3$ & $\underline{\alpha}$

and equating to zero gives to m.l.e.'s denoted by $\hat{\lambda}_1^*, \hat{\lambda}_2^*, \hat{\lambda}_3^*, \hat{\underline{\alpha}}^*$. The likelihood ratio test statistic is

$$\Lambda_3 = \frac{L(\hat{\lambda}_1^*, \hat{\lambda}_2^*, \hat{\lambda}_3^*, \hat{\underline{\alpha}}^*, \underline{0}, \underline{0})}{L(\hat{\lambda}_1, \hat{\lambda}_2, \hat{\lambda}_3, \hat{\underline{\alpha}}, \hat{\underline{\beta}}, \hat{\underline{\gamma}})} \tag{23}$$

If $n$ is large, then $-2\log(\Lambda_3)$ may be compared with a suitable $\chi^2_{2d}$ percentile in order to decide whether $H_0$ should be rejected or not.

### 4.2.4. Testing $H_0 : \underline{\beta} = \underline{0}$

Here, we are interested in testing whether the observed concomitant does not affect the intercept term of the pseudo-Poisson model. Now, under the null hypothesis, the natural parameter space is $\Theta_0 = \{(\lambda_1, \lambda_2, \lambda_3, \underline{\alpha}, \underline{\beta}, \underline{\gamma})^\top : \lambda_1 > 0, \lambda_2 \geq 0, \lambda_3 \geq 0, \underline{\alpha} \in \mathbb{R}^d, \underline{\beta} = \underline{0}, \underline{\gamma} \in \mathbb{R}^d\}$. Again, taking partial derivatives of Equation (4) with respect to each parameters $\lambda_1, \lambda_2, \lambda_3$ & $\underline{\alpha}, \underline{\gamma}$ and equating to zero gives to m.l.e.'s denoted by $\hat{\lambda}_1^*, \hat{\lambda}_2^*, \hat{\lambda}_3^*, \hat{\underline{\alpha}}^*, \hat{\underline{\gamma}}^*$. The likelihood ratio test statistic is

$$\Lambda_4 = \frac{L(\hat{\lambda}_1^*, \hat{\lambda}_2^*, \hat{\lambda}_3^*, \hat{\underline{\alpha}}^*, \underline{0}, \hat{\underline{\gamma}}^*)}{L(\hat{\lambda}_1, \hat{\lambda}_2, \hat{\lambda}_3, \hat{\underline{\alpha}}, \hat{\underline{\beta}}, \hat{\underline{\gamma}})} \tag{24}$$

If $n$ is large, then $-2\log(\Lambda_4)$ may be compared with a suitable $\chi^2_d$ percentile in order to decide whether $H_0$ should be rejected or not.

### 4.2.5. Testing $H_0 : \underline{\gamma} = \underline{0}$

In this case, we wish to determine whether the concomitant does not affect the dependence structure of the pseudo-Poisson model. Thus, under the null hypothesis, parameter space is $\Theta_0 = \{(\lambda_1, \lambda_2, \lambda_3, \underline{\alpha}, \underline{\beta}, \underline{\gamma})^\top : \lambda_1 > 0, \lambda_2 \geq 0, \lambda_3 \geq 0, \underline{\alpha} \in \mathbb{R}^d, \underline{\beta} \in \mathbb{R}^d, \underline{\gamma} = \underline{0}\}$. Now, taking partial derivatives of Equation (4) with respect to the parameters $\lambda_1, \lambda_2, \lambda_3$ & $\underline{\alpha}, \underline{\beta}$ and equating to zero gives, m.l.e.'s denoted by $\hat{\lambda}_1^*, \hat{\lambda}_2^*, \hat{\lambda}_3^*, \hat{\underline{\alpha}}^*, \hat{\underline{\beta}}^*$. The likelihood ratio test statistic is

$$\Lambda_5 = \frac{L(\hat{\lambda}_1^*, \hat{\lambda}_2^*, \hat{\lambda}_3^*, \hat{\underline{\alpha}}^*, \hat{\underline{\beta}}^*, \underline{0})}{L(\hat{\lambda}_1, \hat{\lambda}_2, \hat{\lambda}_3, \hat{\underline{\alpha}}, \hat{\underline{\beta}}, \hat{\underline{\gamma}})} \tag{25}$$

If $n$ is large, then $-2\log(\Lambda_5)$ may be compared with a suitable $\chi^2_d$ percentile in order to decide whether $H_0$ should be rejected or not.

In the next examples, we are interested in testing some hypotheses concerning the relationship between the explanatory and response variables. In particular, we are interested in testing whether the regression planes are parallel or if they are coincident. We illustrate the testing procedure using the simple sub-model given by

$$X_1 \sim \mathscr{P}\left(\exp\left(\sum_{j=1}^d u_{ij}\alpha_j\right)\right) \tag{26}$$

and

$$X_2 | X_1 = x_1 \sim \mathscr{P}\left(\exp\left(\sum_{j=1}^d u_{ij}\gamma_j\right) x_1\right). \tag{27}$$

#### 4.2.6. Testing for Parallelism

In the following, we are interested in testing whether the planes on which the means lie are parallel. If we set $u_{1i} = 1$ for $i \in \{1, \ldots, n\}$ then the two marginal means are

$$log(E(X_1)) = \alpha_1 + \sum_{j=2}^{d} u_{ij}\alpha_j \tag{28}$$

$$log(E(X_2)) = \alpha_1 + \gamma_1 + \sum_{j=2}^{d} u_{ij}(\alpha_j + \gamma_j). \tag{29}$$

For the bivariate pseudo-Poisson regression model specified in (26) and (27), now it is interesting to examine the hypothesis that the planes on which the mean lies are parallel. This is equivalent to testing for the hypothesis $H_0 : \gamma_j = 0$, for $j \in \{1, \ldots, d\}$. Under the null hypothesis, the pseudo-Poisson regression model will be

$$X_1 \sim \mathscr{P}\left( \exp\left( \alpha_1 + \sum_{j=2}^{d} u_{ij}\alpha_j \right) \right) \tag{30}$$

and

$$X_2 | X_1 = x_1 \sim \mathscr{P}\left( \exp(\gamma_1)x_1 \right). \tag{31}$$

The log-likelihood is

$$\begin{aligned}
log(L_{PH}) = & -\sum_{i=1}^{n} \exp\left( \alpha_1 + \sum_{j=2}^{d} u_{ij}\alpha_j \right) + \sum_{i=1}^{n} x_{1i} \log\left( \exp\left( \alpha_1 + \sum_{j=2}^{d} u_{ij}\alpha_j \right) \right) \\
& - \sum_{i=1}^{n} \exp(\gamma_1)x_{1i} + \sum_{i=1}^{n} x_{2i} \log\left( \exp(\gamma_1)x_{1i} \right) - \sum_{i=1}^{n} \log(x_{1i}!x_{2i}!).
\end{aligned} \tag{32}$$

Note that testing for parallelism for the model specified in (30) and (31) is equivalent to testing for the observed concomitant and has no effect on the conditional distribution of $X_2$ given $X_1$. Now, partial differentiation with respect to $\gamma_1$ and $\alpha_j$, $j \in \{1, \ldots, d\}$ and equating to zero gives us

$$log(\bar{X}_1) = \alpha_1 + \sum_{j=2}^{d} u_{ij}\alpha_j$$

$$\sum_{i=1}^{n} x_{1i}u_{ij} = \sum_{i=1}^{n} \exp\left( \alpha_1 + \sum_{j=2}^{d} \alpha_j u_{ij} \right), \; j \in \{2, \ldots, d\} \tag{33}$$

Solving the above $d$ equations leads us to the m.l.e. of $\alpha_i$ denoted by $\alpha_{Pj}^{\hat{*}}$, $j \in \{1, \ldots, d\}$ and the m.l.e. of $\gamma_1$ is

$$\gamma_{P1}^{\hat{*}} = log\left( \sum_{i=1}^{n} x_{2i} \log(x_{1i}) - \sum_{i=1}^{n} x_{1i} \right). \tag{34}$$

Now, we denote the obtained m.l.e.'s under the complete parameter space by $\hat{\alpha}_{Pj}$ and $\hat{\gamma}_{Pj}$, $j \in \{1, \ldots, d\}$. The likelihood ratio test statistic is

$$\Lambda_P = \frac{L_{PH}(\hat{\alpha}_P^{\hat{*}}, \underline{0})}{L_P(\underline{\hat{\alpha}_P}, \underline{\hat{\gamma}_P})},$$

where $L_P(.,.)$ is the likelihood of the model in (26) & (27) and (30) & (31). If $n$ is large, then $-2log(\Lambda_P)$ may be compared with a suitable $\chi^2_{d-1}$ percentile in order to decide whether $H_0$ should be rejected or not.

### 4.2.7. Testing for Coincidence

Here, we assume that the regression relationship does not change from time 1 to time 2 which will occur if the planes on which means lies are coincident. Now, for the given model in (26) and (27), the two marginal means are

$$log(E(X_1)) \quad = \quad \sum_{j=1}^{d} u_{ij}\alpha_j \tag{35}$$

$$log(E(X_2)) \quad = \quad \sum_{j=1}^{d} u_{ij}(\alpha_j + \gamma_j). \tag{36}$$

The assumption of coincidence leads us to test $H_0 : \gamma_j = 0$, for $j \in \{1, \ldots, d\}$. Denote by $\hat{\alpha}^*_{Cj}$ for $j \in \{1, \ldots, d\}$ are m.l.e.'s under the null hypothesis and by $\hat{\alpha}_{Cj}$ and $\hat{\gamma}_{Cj}$ for $j \in \{1, \ldots, d\}$ are m.l.e.'s under complete parameter space, for . Now, the likelihood ratio test statistic is

$$\Lambda_C = \frac{L_{CH}(\hat{\underline{\alpha}^*_P}, \underline{0})}{L_C(\hat{\underline{\alpha}_P}, \hat{\underline{\gamma}_P})},$$

where $L_{CH}(.,)$ and $L_C(.,.)$ are likelihood under null and complete parameter space, respectively. If $n$ is large, then $-2log(\Lambda_C)$ may be compared with a suitable $\chi^2_d$ percentile to decide whether $H_0$ should be rejected or not.

## 5. Applications

In the following two subsections, we illustrate a simulation study and give examples of real-life applications of the bivariate pseudo-Poisson regression model.

### 5.1. Simulation

We have simulated 2000 data sets of sample size $n = 20, 30, 50, 100, 200, 500, 1000$ for the parameter values $\lambda_1 = 1$, $\lambda_2 = 1$, $\lambda_3 = 4$, $\alpha_1 = 1$, $\alpha_2 = 0$, $\alpha_3 = -1$, $\beta_1 = 0$, $\beta_2 = 1$, $\beta_3 = 1$, $\gamma_1 = 0$, $\gamma_2 = 0$ and $\gamma_3 = 1$ from the pseudo-Poisson regression model. We refer to Figures 1–4 for the bootstrapped distribution of each of the parameters. The numerical evidence suggests that as sample size increases, m.l.e.'s approach the true parameter values with standard errors that are decreasing as the sample size increases.

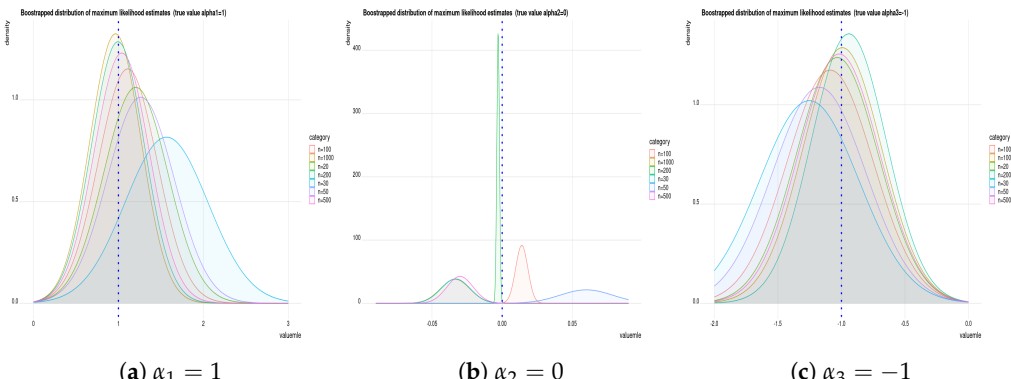

(a) $\alpha_1 = 1$        (b) $\alpha_2 = 0$        (c) $\alpha_3 = -1$

**Figure 1.** Boostrapped distribution of $\underline{\alpha} = (\alpha_1, \alpha_2, \alpha_3)^\top$.

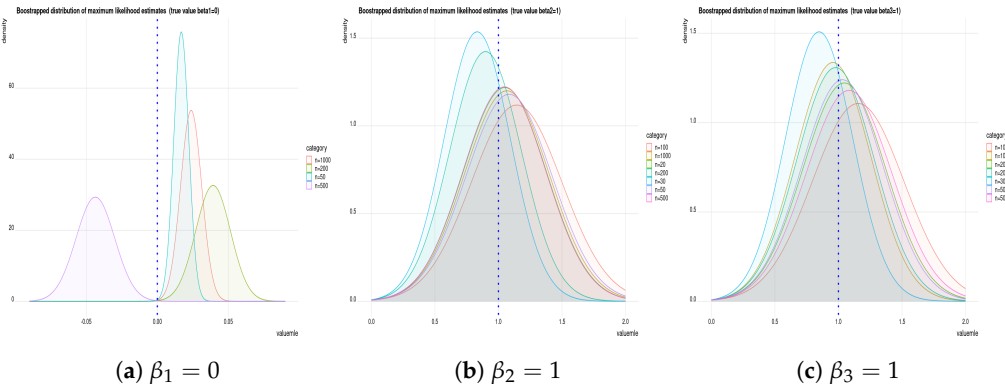

(**a**) $\beta_1 = 0$       (**b**) $\beta_2 = 1$       (**c**) $\beta_3 = 1$

**Figure 2.** Boostrapped distribution of $\underline{\beta} = (\beta_1, \beta_2, \beta_3)^\top$.

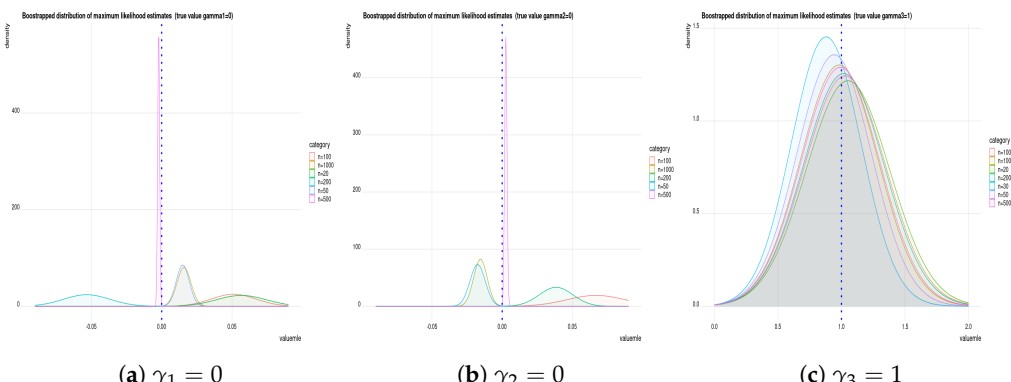

(**a**) $\gamma_1 = 0$       (**b**) $\gamma_2 = 0$       (**c**) $\gamma_3 = 1$

**Figure 3.** Boostrapped distribution of $\underline{\gamma} = (\gamma_1, \gamma_2, \gamma_3)^\top$.

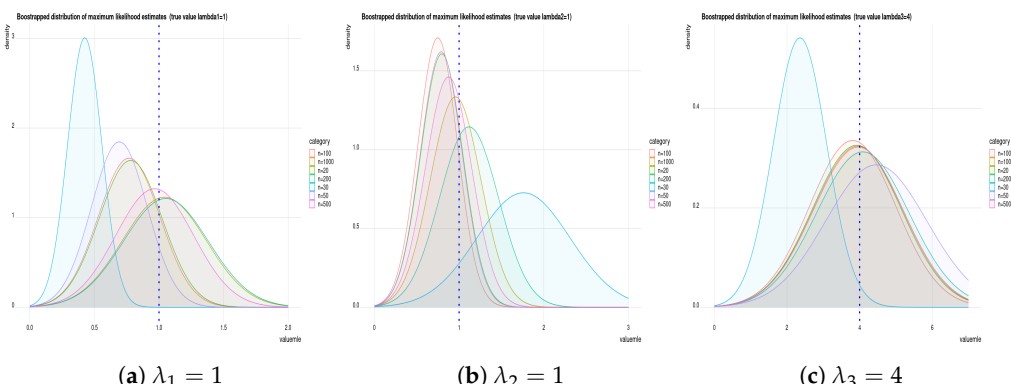

(**a**) $\lambda_1 = 1$       (**b**) $\lambda_2 = 1$       (**c**) $\lambda_3 = 4$

**Figure 4.** Boostrapped distribution of $(\lambda_1, \lambda_2, \lambda_3)$.

*5.2. Real-Life Data*

5.2.1. Australian Health Service Utilization Data: 1977–1978

We consider a data set which is mentioned in Islam and Chowdhury [14] that is part of the Health and Retirement Study (HRS). The data represent the number of conditions ever had ($X_1$) as mentioned by the doctors and utilization of healthcare services (say, hospital, nursing home, doctor and home care) ($X_2$). The concomitant variables are Gender, Age, Hispanic, and Veteran.

The marginal estimated dispersion indices are 0.779 and 1.029. The sample Pearson correlation coefficient between $X_1$ and $X_2$ is 0.063. We can conclude that marginal $X_1$ is approximately equi-dispersed and marginal $X_2$ is slightly over-dispersed. Further, the data were also tested for independence and it was concluded that the assumption was rejected, cf. Arnold and Manjunath [7] pages 2321–2322.

We refer to the Table 1 for the log-likelihood values for the following models:

Full Model: The parameters are $\lambda_1, \lambda_2, \lambda_3, \underline{\alpha} = (\alpha_1, \alpha_2, \alpha_3, \alpha_4)^\top$, $\underline{\beta} = (\beta_1, \beta_2, \beta_3, \beta_4)^\top, \underline{\gamma} = (\gamma_1, \gamma_2, \gamma_3, \gamma_4)^\top$

Mirrored, Model (in which $X_1$ and $X_2$ are interchanged): The parameters are $\lambda_1, \lambda_2, \lambda_3$, $\underline{\alpha} = (\alpha_1, \alpha_2, \alpha_3, \alpha_4)^\top, \underline{\beta} = (\beta_1, \beta_2, \beta_3, \beta_4)^\top, \underline{\gamma} = (\gamma_1, \gamma_2, \gamma_3, \gamma_4)^\top$

Sub-Model I ($\lambda_2 = 0$): The parameters are $\lambda_1, \lambda_3, \underline{\alpha} = (\alpha_1, \alpha_2, \alpha_3, \alpha_4)^\top$, $\underline{\gamma} = (\gamma_1, \gamma_2, \gamma_3, \gamma_4)^\top$

Sub-Model II ($\lambda_2 = \lambda_3$): The parameters are $\lambda_1, \lambda_3, \underline{\alpha} = (\alpha_1, \alpha_2, \alpha_3, \alpha_4)^\top$, $\underline{\gamma} = (\gamma_1, \gamma_2, \gamma_3, \gamma_4)^\top$

Sub-Model II (Mirrored): The parameters are $\lambda_1, \lambda_3, \underline{\alpha} = (\alpha_1, \alpha_2, \alpha_3, \alpha_4)^\top$, $\underline{\gamma} = (\gamma_1, \gamma_2, \gamma_3, \gamma_4)^\top$.

**Table 1.** Models for the Australian Health Service Utilization Data.

| Models | No. Parameters | Log-Likelihood |
|---|---|---|
| Full Model | 15 | −52,654.13 |
| Mirrored, Full Model | 15 | −16,229.24 |
| Sub-Model I ($\lambda_2 = 0$) | 11 | −16,586.07 |
| Sub-Model II ($\lambda_2 = \lambda_3$) | 11 | −16,371.95 |
| Mirrored Sub-Model II ($\lambda_2 = \lambda_3$) | 11 | −17,585.37 |

The mirrored Full Model fits the data best. For the detailed discussion on the mirrored model, see Arnold and Manjunath [7] page 2323. In Islam and Chowdhury [14], page 122, the authors fitted the Poisson–Poisson regression model for the same data set. Note that the Poisson–Poisson regression model is a sub-model of the pseudo-Poisson regression model when $\lambda_2 = 0$. Hence, we conclude that our generalized pseudo-Poisson mirrored model fits the data better than the Poisson–Poisson regression model. The parameter estimates for the pseudo-Poisson mirrored model and their standard errors are displayed in Table 2.

Further, we tested for the significance of the regression coefficients. With reference to Table 3, the computed $-2 \log \lambda$ and compared with $\chi^2$ table values with respective degrees of freedom and the size of 0.05 or 0.10 and concluded that there is not enough evidence to accept the null hypotheses.

**Table 2.** Final model estimates and its standard error (s.e.) for the Australian Health Service Utilization Data.

| Parameter | m.l.e. | s.e. |
|---|---|---|
| $\alpha_1$ | 0.292 | 0.039 |
| $\alpha_2$ | −0.008 | 0.004 |
| $\alpha_3$ | −0.186 | 0.058 |
| $\alpha_4$ | 0.140 | 0.042 |
| $\beta_1$ | −0.132 | 0.0273 |
| $\beta_2$ | 0.016 | 0.0036 |
| $\beta_3$ | 0.038 | 0.0277 |
| $\beta_4$ | 0.053 | 0.035 |
| $\gamma_1$ | 1.636 | 0.656 |
| $\gamma_2$ | −0.025 | 0.039 |
| $\gamma_3$ | −0.996 | − |
| $\gamma_4$ | −0.148 | 0.273 |
| $\lambda_1$ | 1.172 | 0.385 |
| $\lambda_2$ | 0.824 | 0.224 |
| $\lambda_3$ | 0.313 | 1.037 |

**Table 3.** Hypothesis testing for the Australian Health Service Utilization Data.

| Hypothesis | $\log \Lambda^* - \log \Lambda$ | d.f. |
|:---:|:---:|:---:|
| $\underline{\alpha} = \underline{\beta} = \underline{\gamma} = \underline{0}$ | $-113.7227$ | 12 |
| $\underline{\alpha} = \underline{0}$ | $-103.483$ | 4 |
| $\underline{\beta} = \underline{\gamma} = \underline{0}$ | $-28.26604$ | 8 |
| $\underline{\beta} = \underline{0}$ | $-24.26104$ | 4 |
| $\underline{\gamma} = \underline{0}$ | $-6.857175$ | 4 |

### 5.2.2. Road Safety Data

The second data set is on road safety, published by the Department for Transport, United Kingdom. The data comprise information about personal injury road accidents in Great Britain and the consequent casualties on public roads. The concomitant variables are Gender of the driver (Male = 1, Female = 0), Area (Urban = 0, Rural = 1), Accident Severity (Fatal Severity = 1 else = 0), Accident Severity ( Serious Severity = 1, else = 0), and Light condition (Daylight = 1, Others = 0).

We refer to Table 4 for the log-likelihood values for the following:

Full Model: parameters are $\lambda_1, \lambda_2, \lambda_3, \underline{\alpha} = (\alpha_1, \alpha_2, \alpha_3, \alpha_4, \alpha_5)^\top,$
$\underline{\beta} = (\beta_1, \beta_2, \beta_3, \beta_4, \beta_5)^\top, \underline{\gamma} = (\gamma_1, \gamma_2, \gamma_3, \gamma_4, \gamma_5)^\top$

Mirrored, Model ($X_1$ and $X_2$ are interchanged): parameters are $\lambda_1, \lambda_2, \lambda_3,$
$\underline{\alpha} = (\alpha_1, \alpha_2, \alpha_3, \alpha_4, \alpha_5)^\top, \underline{\beta} = (\beta_1, \beta_2, \beta_3, \beta_4, \beta_5)^\top, \underline{\gamma} = (\gamma_1, \gamma_2, \gamma_3, \gamma_4, \gamma_5)^\top$

Sub-Model I ($\lambda_2 = 0$): parameters are $\lambda_1, \lambda_3, \underline{\alpha} = (\alpha_1, \alpha_2, \alpha_3, \alpha_4, \alpha_5)^\top,$
$\underline{\gamma} = (\gamma_1, \gamma_2, \gamma_3, \gamma_4, \gamma_5)^\top$

Sub-Model I (Mirrored): parameters are $\lambda_1, \lambda_3, \underline{\alpha} = (\alpha_1, \alpha_2, \alpha_3, \alpha_4, \alpha_5)^\top,$
$\underline{\gamma} = (\gamma_1, \gamma_2, \gamma_3, \gamma_4, \gamma_5)^\top$

Sub-Model II ($\lambda_2 = \lambda_3$): parameters are $\lambda_1, \lambda_3, \underline{\alpha} = (\alpha_1, \alpha_2, \alpha_3, \alpha_4, \alpha_5)^\top,$
$\underline{\gamma} = (\gamma_1, \gamma_2, \gamma_3, \gamma_4, \gamma_5)^\top$

Sub-Model II (Mirrored): parameters are $\lambda_1, \lambda_3, \underline{\alpha} = (\alpha_1, \alpha_2, \alpha_3, \alpha_4, \alpha_5)^\top,$
$\underline{\gamma} = (\gamma_1, \gamma_2, \gamma_3, \gamma_4, \gamma_5)^\top.$

**Table 4.** Models for the Road safety data.

| Models | No. Parameters | Log-Likelihood |
|:---:|:---:|:---:|
| Full Model | 18 | $-223,743.3$ |
| Mirrored Full Model | 18 | $-243,538.7$ |
| Sub-Model I($\lambda_2 = 0$) | 11 | $-251,937.1$ |
| Mirrored Sub-Model I($\lambda_2 = 0$) | 11 | $-37,599.63$ |
| Sub-Model II($\lambda_2 = \lambda_3$) | 11 | $-36,201.52$ |
| Mirrored Sub-Model II($\lambda_2 = \lambda_3$) | 11 | $-36,516.22$ |

We refer to Table 4 and conclude that the Full Model fits the road safety data and refer to Table 5 for the estimates and their standard errors.

**Table 5.** Final model estimates and its standard error (s.e.) Road safety data.

| Parameter | m.l.e. | s.e. |
|:---:|:---:|:---:|
| $\alpha_1$ | 1.002 | 0.006 |
| $\alpha_2$ | 0.999 | 0.005 |
| $\alpha_3$ | 0.999 | 0.017 |
| $\alpha_4$ | 0.999 | 0.005 |
| $\beta_1$ | 1.000 | 0.005 |
| $\beta_2$ | 1.004 | 0.005 |
| $\beta_3$ | 1.003 | 0.005 |
| $\beta_4$ | 1.000 | 0.015 |
| $\gamma_1$ | 0.999 | 0.0036 |
| $\gamma_2$ | 1.005 | 0.004 |
| $\gamma_3$ | 1.355 | – |
| $\gamma_4$ | 1.319 | – |
| $\lambda_1$ | 1.010 | – |
| $\lambda_2$ | 1.105 | – |
| $\lambda_3$ | $-0.078$ | 0.007 |

## 6. Concluding Remarks

The bivariate pseudo-Poisson model with its straightforward structure with no restrictions on the conditional mean function allows us to model a variety of dependence structures, including positive and negative correlation. Introducing explanatory variables in such models will be a useful additional to the toolkit for modelers dealing with bivariate count data which have positive or negative correlation. In the current note, we explored distributional and inferential aspects of such models and also included a simulation and real-life data applications. We emphasize the advantage of considering the current model over other available count regression models in Section 5.2. The bivariate pseudo-Poisson regression model has a simple structure, straightforward parameter estimation and fast computation, and will deserve a place in the analysis of count data sets with concomitants.

**Author Contributions:** Both authors equally contributed. All authors have read and agreed to the published version of the manuscript.

**Funding:** The second author's research was sponsored by the Institution of Eminence (IoE), University of Hyderabad (UoH-IoE-RC2-21-013).

**Institutional Review Board Statement:** Not applicable.

**Informed Consent Statement:** Not applicable.

**Data Availability Statement:** The data sets used in the current article are available at bpglm: R package for Bivariate Poisson GLM with Covariates.

**Conflicts of Interest:** The authors declare no conflict of interest.

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
