# Peer review of "Pseudo-Poisson Distributions with Concomitant Variables"

_mca, doi:10.3390/mca28010011_

Round 1

Reviewer 1 Report

See scanned report in attachment

Author Response

Dear Sir,

We thank your time and effort in correcting our typos and constructive suggestions.  We are attaching the detailed response to the comments.

Also, we make a note that Section 5.2.3 has been deleted because it is not adding any additional justification to the proposed criteria. 

Reviewer 2 Report

Review of the paper entitled " Pseudo-Poisson distributions with concomitant variables " submitted to MCA

The paper can be summarized as follows: According to Arnold and Manjunath (2021), the bivariate pseudo-Poisson distribution will be the preferred model for bivariate data, with one marginal equidispersed and the other marginal over-dispersed. This is as a result of its straightforward parameter estimates, straightforward structure, and quick calculation. In the current note, we discuss the distributional and inferential features of the augmented models as well as the influence of concurrent variables on the bivariate pseudo-Poisson parameters. A brief simulation study and an illustration of its application to real-world data are also included.

Evaluation: The paper considers a well-known but still interesting statistical subject and develops an interesting methodology based on bivariate pseudo-Poisson distributions. The paper is well written and of good overall quality. However, in my opinion, it lacks substance, and the model comparisons are not fair enough with the existing models in the literature. For these reasons, I encourage the authors to do a revision according to the following points:

o The literature about "bivariate Poisson distribution and regression model" is very large ! It is very surprising to see so few references in the paper on such a well-known subject.
For instance, please see the reference below AND the references therein.

Indranil Ghosh, Filipe Marques & Subrata Chakraborty (2021) A new bivariate Poisson distribution via conditional specification: properties and applications, Journal of Applied Statistics, 48:16, 3025-3047, DOI: 10.1080/02664763.2020.1793307

o Some parts are "too heavy" for nothing. I mainly think of the partial differentiation in Section 4.1. In other words, I believe the paper can be reduced to theory in order to emphasize the model in practice (which is still the primary goal of a statistical model).

o The application is interesting, but there is an evident lack of "model comparison" for the same data set. I mean, the authors compare the performance of the "full model" and various sub-models. But what about the numerous existing bivariate count regression models ? The comparison is too egocentric to really measure the importance of the model. I suggest a comparison of performance with other models. This is important. 
One can consider the regression version of the models in the following reference, for instance:

Indranil Ghosh, Filipe Marques & Subrata Chakraborty (2021) A new bivariate Poisson distribution via conditional specification: properties and applications, Journal of Applied Statistics, 48:16, 3025-3047, DOI: 10.1080/02664763.2020.1793307

Minor: 

o In definition 1, what represents "bold X" and "bold x"? (just bold typo I guess)

Author Response

Dear Sir,

Thank you very much for your constructive comments on our article.

I have considered them and attaching the detailed response to it. 

Also, we make a note that Section 5.2.3 has been deleted because it is not adding any additional justification to the proposed criteria. 
